# **Unveiling the Limits of Deep Learning Models in Hydrological Extrapolation Tasks**

Sanika Baste<sup>1</sup>, Daniel Klotz<sup>2, 3</sup>, Eduardo Acuña Espinoza<sup>1</sup>, Andras Bardossy<sup>4</sup>, and Ralf Loritz<sup>1</sup>

<sup>1</sup>Institute of Water and Environment, Karlsruhe Institute of Technology (KIT), Karlsruhe, Germany

**Correspondence:** Sanika Baste (sanika.baste@kit.edu)

Abstract. Long Short-Term Memory (LSTM) networks have shown strong performance in rainfall-runoff modelling, often surpassing conventional hydrological models in benchmark studies. However, recent studies raise questions about their ability to extrapolate, particularly under extreme conditions that exceed the range of their training data. This study examines the performance of a stand-alone LSTM trained on 196 catchments in Switzerland when subjected to synthetic design precipitation events of increasing intensity and varying duration. The model's response is compared to that of a hybrid model and evaluated against hydrological process understanding. Our study reiterates that the stand-alone LSTM is not capable of predicting discharge values above a theoretical limit (which we have calculated for this study to be 73 mm d<sup>-1</sup>), and we show that this limit is below the range of the data the model was trained on (183 mm d<sup>-1</sup> when trained on CAMELS-CH). Furthermore, the LSTM exhibits a concave runoff response under extreme precipitation, indicating that event runoff coefficients decrease with increasing design precipitation—a phenomenon not observed in the hybrid model used as a benchmark. We show that saturation of the LSTM cell states alone does not fully account for this characteristic behavior, as the LSTM does not reach full saturation, particularly for the 1-day events. Instead, its gating structures prevent new information about the current extreme precipitation from being incorporated into the cell states. Adjusting the LSTM architecture, for instance, by increasing the number of hidden states, and/or using a larger, more diverse training dataset can help mitigate the problem. However, these adjustments do not guarantee improved extrapolation performance, and the LSTM continues to predict values below the range of the training data or show unfeasible runoff responses during the 1-day design experiments. Despite these shortcomings, our findings highlight the inherent potential of stand-alone LSTMs to capture complex hydro-meteorological relationships. We argue that more robust training strategies and model configurations could address the observed limitations, preserving the promise of stand-alone LSTMs for rainfall-runoff modelling.

# 20 1 Introduction

Deep learning models, particularly Long Short-Term Memory (LSTM; Hochreiter and Schmidhuber, 1997) networks, have become important tools in rainfall–runoff modelling. The current prototypical setup was introduced by Kratzert et al. (2019a), who trained a single LSTM model for 531 basins across the United States (and achieved superior performance compared to

<sup>&</sup>lt;sup>2</sup>Interdisciplinary Transformation University Austria, Linz, Austria

<sup>&</sup>lt;sup>3</sup>Google Research, Vienna, Austria

<sup>&</sup>lt;sup>4</sup>Institut für Wasser- und Umweltsystemmodellierung, Universität Stuttgart, Stuttgart, Germany

several traditional process-based models). Similar results were confirmed in follow-up work, such as the study by Lees et al. (2021) in Great Britain or Loritz et al. (2024) in Germany. However, as with any model, certain best practices for setting up LSTM-based models are essential to achieve good predictive performance. Among the most important, is training the LSTMs on large, comprehensive, and diverse datasets (Kratzert et al., 2024)—such as Catchment Attributes and Meteorology for Large-sample Studies (CAMELS-US; Addor et al., 2017; Newman et al., 2015).

A behavior that LSTMs exhibit, is that their states can saturate when they ingest new inputs. The mechanism that leads to this behavior is the use of hyperbolic tangent (tanh) and sigmoid activation functions inside LSTM cell. These saturate when the output approaches their asymptotic extremes (Chen and Chang, 1996; Rakitianskaia and Engelbrecht, 2015). Kratzert et al. (2024) identified the saturation of the tanh function in the computation of the hidden states ( $h_t = o_t \odot \tanh(c_t)$ ), where  $c_t$  are the cell states and  $o_t$  is the output gate; Appendix A1) as a key factor that limits the ability of the LSTMs to predict extreme discharge values. As  $c_t$  grows tanh caps them, restricting the transmission of meaningful information, such as meteorological forcing signals. The severity of this saturation effect depends on the learned weights and biases, and hence on the range and diversity of the training data. In hydrological modelling, the circumstance that model predictions are restricted to the empirical support of the data is unsatisfactory—particularly for the prediction of extremes, which is a key modelling aspect. Considering the rapid rise in the application of LSTMs and other deep learning models in rainfall—runoff modelling, we believe that a deeper understanding of their current limitations is essential. This study therefore aims to examine the extrapolation behavior of LSTMs to extreme rainfall—runoff events that lie outside the range of the training data. Albeit the term "extrapolation" is difficult to pinpoint technically—especially in the context of high-dimensional datasets and deep learning models (Balestriero et al., 2021)—the events that we consider in our study are by construction either at the edge of, or outside the range of the observed data (with regard to precipitation).

Previous studies (e.g., Frame et al., 2022; Acuña Espinoza et al., 2024a; Song et al., 2024) have explored the predictive accuracy of LSTMs in extreme runoff scenarios by adopting training/test splits that deliberately exclude certain high-flow values during training. In a stress test setting, Frame et al. (2022) found that, when compared with two conceptual hydrological models, a stand-alone LSTM outperformed one of the former for the most extreme rainfall—runoff events in the CAMELS-US, and was only slightly worse than the second. Acuna Espinoza et al. (2024b) used the same setting to demonstrate that a hybrid model, combining a conceptual hydrological model with an LSTM, was slightly better than a stand-alone LSTM at predicting the most extreme events in the CAMELS-US dataset. In the study, the stand-alone LSTM performed particularly well for the overall evaluation, but for the most extreme events, the LSTM's response showed major deviations from the hybrid model and a conceptual model—exhibiting a distribution of simulated extreme values with no tail (see Fig. 5(a) in Acuna Espinoza et al. (2024b)). On the other hand, Song et al. (2024) (in a slightly different setting) found that a hybrid model, similar to the one used in Acuna Espinoza et al. (2024b) outperformed the stand-alone LSTM. The stand-alone LSTM, the mass-conserving LSTM (MC-LSTM in Frame et al., 2022), and hybrid models performed similarly when evaluated using standard metrics; however, the studies provided notably different interpretations regarding whether, and to what extent, LSTMs can successfully

extrapolate to extreme events.

60

Although the stress tests in Frame et al. (2022); Acuna Espinoza et al. (2024b) systematically test the model's ability to handle increasingly extreme events, it is not realistic from a practical perspective. In real-world applications, modellers would not intentionally exclude known extremes from their training datasets, particularly when using data-driven models. In this study, we propose a complementary approach for investigation: Rather than withholding extreme events during training, we force the LSTM with design precipitation values (as commonly used in infrastructure planning and engineering; Global Water Partnership (GWP) and World Meteorological Organization (WMO), 2013). These precipitation values, which are derived using statistical models, can exceed historical observations, but are considered physically plausible (World Meteorological Organization (WMO), 1973, 2009). This allows us to probe the model's extrapolation capabilities without imposing artificial constraints on the training data. An intrinsic limitation of our approach is that our augmentation destroys the covariate-structure of the inputs. Hence, in theory, we cannot directly disentangle the effect of the general LSTM out-of-distribution behavior and the one introduced by an actual extreme event of the same kind. This restricts us to a certain coarseness of the analytical depth of our study. However, we argue that the pattern that emerges from our experiments is so clear that it is indicative for the extrapolation behavior of LSTMs in hydrology. Specifically, we compare the LSTM's output with that of a mass-conserving hybrid model (Feng et al., 2022) and assess how both models respond under unprecedented forcing conditions to evaluate the physical realism of the LSTM's predictions.

This study addresses the following research questions:

- 1. Can LSTMs extrapolate to discharge values beyond the training distribution when forced with statistically derived design precipitation events?
- 2. Is the saturation of LSTM memory states the primary reason, which limits their ability to extrapolate to extreme and unprecedented hydrological conditions?
  - 3. How do the inherent assumptions and structural characteristics (inductive biases) of LSTMs influence their ability to simulate realistic hydrological responses under conditions that exceed observed training ranges?

The paper is structured as follows: we give a description of the datasets and the models in Section 2. This section also details out the set-up for the design precipitation experiments and the methodology for calculating saturation in the LSTM network. This is followed by Section 3, where we present the overall model performance and a comparison of model simulations from our design experiments. We discuss the findings and their implications with regard to the three research questions in Section 4 and give our conclusion in Section 5.

#### 2 Data and Methods

In this section, we describe the CAMELS-CH dataset (Section 2.1) and the CAMELS-US dataset (Section 2.2) used for model training and testing. The subsequent subsections (Section 2.3 and Section 2.4) briefly describe the LSTM networks, the hybrid model, and their respective model configurations employed in this study. Following these, the section 2.5 details out the selection of catchments and experimental setup for the design precipitation events. Finally, Section 2.6 explains how we estimate network saturation in the LSTM.

#### 95 2.1 The CAMELS-CH Dataset

The CAMELS-CH dataset (Höge et al., 2023) provides daily hydro-meteorological time series data for 331 basins within Switzerland and neighboring countries, along with static catchment attributes which include topographic, climate, hydrology, soil, land cover, geology, glacier, hydrogeology, and human influence attributes. Due to its diverse topography and climate, Switzerland is often referred to as the 'water tower of Europe' (Höge et al., 2023) and despite its small size, it exhibits significant hydrological variability across different regions. CAMELS-CH includes data for 298 river catchments and 33 lakes. The available data spans from 1 January 1981 to 31 December 2020. In this study, we exclude the lakes and 102 river catchments belonging to France, Germany, Austria, and Italy and focus only on the 196 catchments in Switzerland. From this subset, we exclude another four catchments where preliminary model simulations had negative Nash-Sutcliffe efficiency (NSE). For the CAMELS-CH dataset, the maximum precipitation during the training period is 234 mm d<sup>-1</sup> and was recorded for the Krummbach stream located in southern Switzerland. The maximum observed specific discharge is 183 mm d<sup>-1</sup> which occurred during a flood in the Chli Schliere stream in the Alpnach village in central Switzerland triggered by torrential rains in August 2005 (Federal Department for the Environment and DETEC, 2005).

# 2.2 The CAMELS-US Dataset

105

We use a subset of 531 catchments from the CAMELS-US dataset, which was originally identified by Newman et al. (2015).

This provides daily meteorological forcing from three data sets, Daymet, Maurer, and NLDAS, and daily stream flow measurements from the United States Geological Survey (USGS) spanning from 1980 to 2015. Catchment topographical characteristics, climate and hydrological indices, and soil, land-cover and geological characteristics are also provided. The maximum observed specific discharge for this training dataset is 299 mm d<sup>-1</sup>, which is recorded for the Medina river in Texas. The precipitation observed in Krummbach stream (234 mm d<sup>-1</sup>) in Switzerland is also the maximum precipitation for this combined training dataset.

# 2.3 LSTM model

The hyperparameters of our LSTM network (see Table 1) are guided by the work of Lees et al. (2021) and Acuña Espinoza et al. (2024a) and the model implementation is done using PyTorch (Paszke et al., 2019). We train an ensemble of 5 LSTMs, all with a single layer of 64 nodes, to account for random initialization and stochasticity in the network optimization algorithm.

Table 1. Hyperparameters for LSTM network and hybrid model ensemble

| Hyperparameter           |                                     | Value                |  |
|--------------------------|-------------------------------------|----------------------|--|
|                          | LSTM                                | hybrid Model         |  |
| Number of layers         |                                     | 1                    |  |
| Number of nodes          |                                     | 64                   |  |
| Dropout rate             |                                     | 0.4                  |  |
| Initial forget gate bias |                                     | 3                    |  |
| Initial learning rate    |                                     | 0.001                |  |
| Sequence length          | 365                                 | 730                  |  |
| Batch size               |                                     | 256                  |  |
| No. of epochs            |                                     | 20                   |  |
| Training period          | 1 October 1995                      | to 30 September 2005 |  |
| Test period              | 1 October 2010 to 30 September 2015 |                      |  |

The head-layer for our LSTMs is a fully connected linear layer with a dropout rate of 0.4. We use a batch size of 256 and a sequence length of 365 days for training our LSTMs for a total of 20 epochs. We use a learning rate of  $1 \times 10^{-3}$  for the first ten epochs, and  $5 \times 10^{-3}$  for the remaining ten epochs. The basin averaged Nash-Sutcliffe efficiency (NSE\*) proposed by Kratzert et al. (2019a) is used as a loss function and the algorithm for optimization is ADAM (Kingma and Ba, 2017). We refer the reader to Kratzert et al. (2019a) for a detailed description of the LSTM architecture and about specific details as to how it is typically applied in hydrology. For easy reference, we present the equations describing the forward pass of the LSTM in Appendix A1. The training and testing periods, as mentioned in Table 1, span from October 1995 to September 2005 and October 2010 to September 2015 respectively. For models trained on the CAMELS-CH datasset alone, 5 dynamic forcing variables, precipitation (mm  $d^{-1}$ ), minimum and maximum temperature (°C), relative sunshine duration (%) and snow water equivalent (mm d<sup>-1</sup>) and 22 static catchment attributes (see Appendix A2) form the model input, and we trained the models to target specific discharge (mm d<sup>-1</sup>). While training the LSTM ensemble on CAMELS-CH and CAMELS-US datasets together, 130 we reduce the number of dynamic and static inputs for similarity within the inputs for catchments belonging to the two datasets. For this ensemble, we use only 3 dynamic forcing variables — precipitation (mm  $d^{-1}$ ), minimum and maximum temperature (°C) from the CAMELS-CH and from the Daymet meteorological forcing data of the CAMELS-US — and 12 static catchment characteristics (listed in Appendix A2) from both the datasets as inputs and the daily stream flow data as the target.

# 135 2.4 The hybrid model

We use a type of hybrid model introduced by Feng et al. (2022). The hybrid model uses a modified version of the Hydrologiska Byråns Vattenbalansavdelning (HBV) model (Bergström, 1976, 1992; Aghakouchak and Habib, 2010; Seibert and Vis, 2012; Beck et al., 2020) as a backbone conceptual model. Differentiable parameter learning (dPL) using a single LSTM is used to parameterize a number of modified HBVs. The discharge signal produced by the modified HBVs is averaged and routed

through a unit hydrograph, which produces the final simulated discharge. We implement the  $\delta_n(\beta^t, \gamma^t)$  version of the hybrid model with a collection of 16 modified HBV models with dynamic parameterization. A detailed description of this model can be found in Feng et al. (2022). While the stand-alone LSTM produces specific discharge as the output, in the hybrid model, the LSTM produces as many outputs as is the number of parameters required by 16 HBVs and the unit hydrograph routing. In our hybrid model, the LSTM estimates 210 model parameter at each time step (13 HBV parameters\*16 HBV models+2 routing 145 parameters). The hyperparameters of the LSTM component and in the hybrid model, and the data split implemented for training and testing are described in Table 1. The hybrid model receives a sequence length of 730 days, the first 365 values from which are used to initialize the internal states of the HBV models (warm-up period) and do not contribute to loss calculation. We choose to train the two models with different sequence lengths, because we wish to implement the models consistent with methodologies presented in studies by Kratzert et al. (2019a) and Acuna Espinoza et al. (2024b). Thus, we train the LSTM 150 using a seq-to-one approach with a sequence length of 365 and the hybrid model with a seq-to-seq approach and sequence length of 730. Please note that increasing the sequence length of the LSTM to 730 does not increase the model performance. The static and dynamic inputs to the hybrid model are given in Appendix A2. The LSTM component, which parameterizes the conceptual part within the hybrid model, uses the same 5 dynamic and 22 static inputs as the stand-alone LSTM. However, an additional input, potential evapotranspiration (pet sim (mm  $d^{-1}$ )) is explicit to the HBV component therein. Training the stand-alone LSTM with this additional dynamic input, for the sake of similarity in over-all inputs, is redundant, since pet\_sim 155 is computed using temperature and radiation data via the Penman-Monteith equation in CAMELS-CH. When we trained an LSTM ensemble with an additional dynamic input pet sim, it did not change our results. The daily time series for pet sim (mm d<sup>-1</sup>) is obtained from the simulation based hydrometeorological time series of the CAMELS-CH dataset. The optimizer and learning rate schedule is the same for both the models.

#### 160 2.5 Design Precipitation Events: Selection and Experimental Set-up

In this study, we use design precipitation values from an extreme value analysis published by the Federal Office of Meteorology and Climatology (MeteoSwiss; MeteoSwiss, 2022). This includes 1- to 5-day precipitation analyses with annual return interval (ARI) from 1 to 300 years at more than 300 meteorological observation stations. Given that the design precipitation values are only valid on the exact location of the stations (Frei and Fukutome, 2022), we identified a smaller subset of 25 CAMELS-CH catchments that have a meteorological observation station within or at a distance of 2.5 km from the catchment boundary. We acknowledge that, given the diversity in terrain and elevation in Switzerland, and its small-scale spatial climate patterns, access to sophisticated tools enabling better interpolation (Bárdossy and Pegram, 2013) of the extreme values would be ideal. However, due to the lack of such methods and the explicit admission of added uncertainty in the related documentation (Frei and Fukutome, 2022), we proceed with the chosen subset of catchments. The models in our study are trained on catchment-averaged precipitation values but tested using point-scale data, which may introduce inconsistencies and serve as a potential source of error. Nonetheless, given the exploratory nature of our objectives, it is less critical that the exact magnitude of extreme precipitation is captured, as long as the values are physically plausible and reflect regionally extreme conditions. We consider this assumption acceptable for our experimental design, which aims to explore the limitations of LSTM-based

hydrological simulations rather than to support infrastructure planning or flood defense design.

To systematically analyze the simulations of our models in extreme scenarios, we force our models with precipitation events of varying ARI during the test period. For each of the above-mentioned 25 catchments, we identified dates, where the observed precipitation value (mm  $d^{-1}$ ) belonged to the top 99.5th percentile of the distribution of precipitation values during the test period in the respective catchment. A total of 201 events/dates distributed among the 25 test catchments were identified and form a part of subsequent experimental set-up. The minimum replaced precipitation is  $34 \text{ mm d}^{-1}$  and the maximum is 139mm d<sup>-1</sup>. We replaced these by the 1-, 3-, and 5-day design precipitation values with ARI of 50, 100, and 300 years. In the case of 3- and 5-day values, the precipitation volume was distributed uniformly over three and five days, respectively, centered around the identified dates. The LSTM and hybrid model then received this synthetic input for discharge simulations. This approach allows us to test the impact of extreme, but physically plausible, magnitudes of precipitation input for the LSTMbased discharge simulations, under different initial conditions. Our experimental set-up is constrained by the fact that we only manipulate precipitation. Given that other meteorological variables, such as temperature or radiation, are not fully independent of precipitation, our approach does not account for the complex correlation among climate inputs. However, by replacing precipitation values only at times when observed extremes had already occurred, we try to minimize inconsistencies in other meteorological inputs. While this approach has its limitations, it provides a controlled setting to examine how the LSTM and hybrid models respond to unprecedented precipitation magnitudes and reflects to a certain degree a classical hydrological use case, which is, the design of infrastructure.

## 2.6 Measuring saturation in the LSTM

Although saturation can occur at any tanh or sigmoid activation within an LSTM, we focus on the saturation that arises during the computation of the hidden state (the second term in Eq. (A6) in Appendix A1) as discussed by Kratzert et al. (2024). Defining a precise threshold for when tanh saturates is challenging due to its continuous nature. However, previous studies have noted that the useful (non-saturated) region extends until approximately 90% of the saturation level (Chen and Chang, 1996). We hence identify saturation in the said activation when the absolute of its output equals or exceeds 0.9. We define network saturation as the total number of saturated activations (out of the 64 units in the hidden layer). In the following, we will use the term "cell state saturation" to refer specifically to the saturation of the tanh activation function when computing hidden states  $(h_t = \tanh(c_t) \cdot o_t)$ .

#### 3 Results






#### 3.1 LSTM and hybrid model performance

Fig. 1 presents the test performance of the LSTM and hybrid model ensemble as a cumulative distribution function (CDF) of individual catchment performance measured by the NSE (panel (a)). The models' testing is spatially in-sample but temporally

out-of-sample, which means that the models are tested using the same 196 catchments used during the training process, but 205 in a different test period (gauged simulations). The average median NSE achieved by the LSTM ensemble is 0.84 while that for the hybrid model ensemble is slightly lower at 0.79. Both models perform better than the PREVAH model (Viviroli et al., 2009) (median NSE = 0.50 (see Fig. B1)), simulated discharge time series from which are provided with the CAMELS-CH dataset. It is worth noting that the hybrid model performed similarly to the LSTM ensemble in studies by Feng et al. (2022) 210 and Acuna Espinoza et al. (2024b) for the CAMELS-US dataset. However, in this study, we could not replicate the same performance, despite using the exact same model setup and training procedure, possibly because we train and test our models on catchments belonging to the CAMELS-CH dataset. Our investigations did not reveal a specific cause for the slightly lower NSE of the hybrid model. Interestingly, in four specific catchments where the hybrid model exhibited a pronounced drop in performance compared to the LSTM ensemble, the hybrid accurately predicted timing patterns (high correlation) but showed an increasing bias over the duration of the test period. This suggests larger mass balance errors in these catchments that could not be corrected due to the hybrid model's mass-conserving structure. Given that the hybrid model primarily serves as a benchmark for the LSTM ensemble, the observed difference in the global NSE is considered negligible for the objectives of this study. This difference in the global performance of the two models is also true for the subset of the 25 catchments (see Section 2.5) identified for the design experiments.

A comparison of the two model ensembles based on the High Flow Bias (FHV), fraction of missed peaks and peak mean absolute percentage error (MAPE) is shown in panels (b), (c) and (d) of Fig. 1 respectively. The FHV represents the peak flow bias of the flow duration curves for the observed and simulated discharge. The fraction of missed peaks represents the peaks in the observed data that are missed in the simulation. The MAPE is the absolute percentage error for observed peaks and their respective simulated values. All discharge values belonging to the top 2% of the observed (or simlated) distribution are considered as peak values for the calculation of the fraction of missed peaks and MAPE (or FHV). Both model ensembles show similar distribution of FHV and fraction of missed peaks across all catchments. The hybrid model, however, has a higher median MAPE and in general shows greater error associated with peaks. For the 201 events identified in Section 2.5, we calculated the root-mean-squared error (RMSE) of the two model ensembles when they were tested for the observed test dataset (without any synthetic precipitation input). The LSTM ensemble has an RMSE of 1.08 mm d<sup>-1</sup> while the hybrid ensemble has a slightly higher RMSE of 1.22 mm d<sup>-1</sup>.

# 3.2 Theoretical prediction limit and maximum simulated value of the LSTM ensemble





Kratzert et al. (2024) discuss the existence of a theoretical prediction limit (TPL) for a trained LSTM network and provide a mathematical derivation (Appendix C in Kratzert et al., 2024). This theoretical prediction limit depends on the learnable parameters (weights and biases) of the linear head layer that maps the LSTM's hidden states to a single output value. For our LSTM ensemble, the mean theoretical prediction limit is 73 mm d<sup>-1</sup>. This limit means that under no circumstances can the stand-alone LSTM produce a simulated discharge higher than 73 mm d<sup>-1</sup>. This theoretical prediction limit is notably smaller than the maximum specific discharge observed during the training period, about 183 mm d<sup>-1</sup>, which occurred during a flood in the Chli Schliere stream, located in central Switzerland. In total, there are 66 days in the training period during which discharge

**Figure 1.** Cumulative Density Function (CDF) showing the (a) NSE, (b) High Flow Bias (FHV), (c) fraction of missed peaks and (d) Peak Mean Absolute Percentage Error (MAPE) of the LSTM and hybrid model ensemble tested on 196 CAMELS-CH catchments during the test period from 01.10.2010 to 30.09.2015. The solid line represents the mean of the ensemble, and the shaded region depicts the variation within the ensemble. The average median NSE achieved by the LSTM network ensemble is 0.84, while that for the hybrid model ensemble is 0.79.

values exceed 73 mm  $d^{-1},$  representing approximately 0.01% of the total training data.



Our design experiments revealed that the maximum simulated discharge value from the LSTM ensemble is not the theoretical limit of 73 mm d<sup>-1</sup>, but 60 mm d<sup>-1</sup>. This maximum was reached during a 1-day design precipitation event, which had a total precipitation volume of 304 mm, in the Magliaso-Ponte catchment located in southern Switzerland. To further investigate how closely the stand-alone LSTM can approach its theoretical maximum, we tested scenarios with extremely high precipitation intensities up to 1000 mm d<sup>-1</sup> sustained over 3- and 5-day durations. Such values exceed realistic conditions by far, especially considering the fact that the highest total annual precipitation recorded in Switzerland is 4173 mm a<sup>-1</sup> (MeteoSwiss, 2024).

Even under these extreme forcing conditions, the model did not produce a discharge value beyond  $60 \text{ mm d}^{-1}$ . We hence refer to this simulated maximum as the "design limit" of the LSTM. The "design limit" being smaller than the theoretical prediction limit, can be understood as a consequence of not all linear head-layer units contributing fully to the final output.







Training LSTMs with a higher number of hidden states and on a larger, more diverse dataset (as recommended in Kratzert et al., 2024) can raise the theoretical limit, but does not necessarily affect the "design limit" to the same degree. For instance, a single LSTM network with 256 hidden states, compared to one with 64 hidden states, trained on the CAMELS-CH dataset, demonstrates a theoretical prediction limit of 120 mm d<sup>-1</sup>. The "design limit" also increased to 75 mm d<sup>-1</sup>. Similarly, a single LSTM with 256 hidden states, trained on both the CAMELS-CH and CAMELS-US datasets together, achieves a theoretical prediction limit of 194 mm d<sup>-1</sup> and a raised "design limit" of 110 mm d<sup>-1</sup>. Despite the substantial improvements in theoretical prediction limits, the "design limits" remain significantly lower than the maximum discharges encountered during training: 299 mm d<sup>-1</sup> in CAMELS-US and 183 mm d<sup>-1</sup> in CAMELS-CH. While the theoretical limit reflects the maximum potential output based on model parameters, the "design limit" is constrained by the interplay of network weights and activations during inference. Thus, increasing the theoretical maximum by expanding the number of hidden states does not necessarily translate to a higher "design limit".

In contrast, the hybrid model used in our experiments does not exhibit a theoretical limit to discharge predictions. The highest simulated value observed was 144 mm  $d^{-1}$ , which is still lower than the maximum discharge seen during training. However, when forced with increased precipitation, the model's outputs scale more or less linearly with the forcing, demonstrating greater flexibility than the standalone LSTM. Panels (a)-(c) in Fig. 2 show the evolution in the simulated specific discharge for three catchments for a particular, catchment-specific, 1-day design precipitation event with varying ARI from 50 to 300 years. We highlight these three events, as they have the highest runoff generation among the 201 events from the 25 catchments and most clearly exhibit the limiting behavior of the LSTM. Notably, the maximum simulated discharge by the stand-alone LSTM ensemble increase only marginally from ARI 50-year to ARI 300-year in all three catchments. For these events the simulations increase on average by 6% from ARI 50 to ARI 300 years, in contrast to the precipitation, that increases by 39%. The maximum simulated values of these three catchments, which are 48 mm  $d^{-1}$ , 43 mm  $d^{-1}$ , and 60 mm  $d^{-1}$  respectively, are well below the theoretical limit of the LSTM ensemble, but close to the "design limit". From a hydrological viewpoint, this entails that, although rainfall increases significantly, the LSTM simulations have decreasing runoff coefficients. In contrast, we typically observe an increase in runoff coefficients with increasing intensity of extreme events, as increasing area of a catchment becomes saturated (Beven et al., 2021). The hybrid model ensemble on the other hand responds considerably more to the increasing precipitation input, and there is an increase of 51% from ARI 50 to ARI 300 years. The identified patterns in the three events shown in Fig. 2 are also true for the events with the highest runoff generation in each of the 25 test catchments. Such events are specifically important, because they are more likely to push the LSTM to its simulation limits and display the saturation effect. While the precipitation increases by 43% from ARI 50 to ARI 300, the LSTM simulations show an average increase of 25%. Whereas, the hybrid simulations increase by 48%. For the rest of the design events, as runoff generation

**Figure 2.** Evolution of LSTM and hybrid model ensemble simulation for three, catchment specific, 1-day events with increasing ARI for gauges located at (a)Andermatt, (b)Pollegio-Campagna and (c)Magliaso-Ponte and their respective hydrographs (d)-(f). The LSTM ensemble doesn't simulate discharge higher than its theoretical prediction limit (panels (d)-(f)). The increase in the hybrid model simulation is more consistent with hydrological expectation than the LSTM (panels (a)-(c)).

varies depending on the state of the catchment, saturation behavior may or may not be observed as starkly. In catchments with particularly low rainfall-runoff generation, the LSTM ensemble often produces higher runoff estimates than the hybrid model. In such cases, the saturation in LSTM runoff generation is not pronounced either. The closer the LSTM estimates approach the "design limit", the greater is the difference between the hybrid model and the LSTM simulation.



Fig. 3 shows the results of a 3-day (panels (a), (c)) and a 5-day (panels (b), (d)) event at the Magliaso-Ponte gauge, one of the test catchments exhibiting the most pronounced runoff responses. Consistent with observations from the 1-day events, the LSTM network simulations reveal certain characteristic limitations. Nonetheless, for both the 3-day and 5-day events, the hybrid model's peak discharge simulations increase with higher ARIs (see panels (a) for the 3-day event and (b) for the 5-day event in Fig. 3). For most of the test catchments, the stand-alone LSTM response shows similar patterns. But the discrepancy between the hybrid and the LSTM simulations is much smaller for the 3-day events compared to the 1-day events, and even further reduced for the 5-day events.

**Figure 3.** Evolution of LSTM and hybrid model ensemble simulation for gauge located at Magliaso-Ponte for a (a)3-day event and a (b)5-day event with their respective hydrographs (c) and (d).

**Table 2.** Number of nodes (out of 64) of the LSTM network such that output of the  $|\tanh(c_n)| \ge 0.90$ . Ensemble maximum (ensemble minimum) values are reported for single events in each catchment. Due to poor reliability of 5-day extreme precipitation analyses for Andermatt (MeteoSwiss, 2022), the corresponding results are not reported here.

| ID   | Gauge Name        | Event Date | Number of Saturated Nodes |                       |        |        |        |        |        |        |        |
|------|-------------------|------------|---------------------------|-----------------------|--------|--------|--------|--------|--------|--------|--------|
|      |                   |            |                           | Design Experiment ARI |        |        |        |        |        |        |        |
|      |                   |            |                           | 50y                   |        |        | 100y   |        |        | 300y   |        |
|      |                   |            | 1d                        | 3d                    | 5d     | 1d     | 3d     | 5d     | 1d     | 3d     | 5d     |
| 2087 | Andermatt         | 08.08.2013 | 37(28)                    | 45(42)                | -      | 35(27) | 46(43) | -      | 34(26) | 45(43) | -      |
| 2494 | Pollegio-Campagna | 22.05.2014 | 32(26)                    | 51(42)                | 50(44) | 32(26) | 52(39) | 50(45) | 32(26) | 50(40) | 51(45) |
| 2461 | Magliaso-Ponte    | 11.10.2014 | 48(40)                    | 50(41)                | 47(41) | 48(40) | 51(42) | 49(42) | 48(37) | 51(44) | 51(43) |

#### 3.3 Evolution of saturation in the LSTM ensemble

For the events identified in Section 2.5, on average, at least 19% and at most 58% network saturation is observed for precipitation input within the test dataset, meaning without the input of synthetic extreme precipitation. This shall serve as a baseline

to observe how much the network further saturates when subject to the synthetic precipitation data during the design events. Table 2 shows the maximum (and minimum) number of saturated LSTM cells (out of 64) for three test catchments across various design events. Notably, in none of the cases do the LSTM's cell states fully saturate. For the 1-day events, on average, the maximum saturation across the ensemble ranged from about 50% to 75%, while the minimum ranged from approximately 41% to 63%. Interestingly, this degree of saturation remained nearly unchanged even as the ARI increased, and the associated precipitation became more intense. Even pushing the model with a very high 1-day precipitation of  $1000 \text{ mm d}^{-1}$  did not cause the cell states to approach complete saturation.

A different pattern emerged, however, when we examined longer-duration events. For the 3-day events, we observed a substantial increase in cell state saturation. This indicates that some cells require more than a single day to accumulate sufficient input signals to reach higher saturation levels. This is thereby controlled by the input and forget gates in an LSTM (Eqs. (A1) and (A2) in Appendix A1). The input gate controls how much new information enters the cell state, while the forget gate determines how much past information is retained or discarded. Over multiple days, the continued influx of rainfall data (regulated 310 by the input gate) and the retention of previously encoded information (controlled by the forget gate) allow the cell states to build up more gradually. With this prolonged input, more cell states move closer to saturation. For the 5-day events, saturation did not increase further, which at first seems contradictory. However, the total precipitation of the 5-day events does not greatly exceed that of the 3-day events. Since the rainfall is spread uniformly over a longer period, it results in a lower daily precipitation intensity. Without sufficiently large daily inputs, the cell states do not accumulate to higher saturation levels, even 315 over multiple days. Thus, while longer durations can facilitate higher saturation when daily precipitation is intense, simply extending the time frame without maintaining high-intensity input does not necessarily lead to further saturation. The number of saturated cell states, hence, provides useful insights. However, the saturation of the cell states is not the only kind of saturation that limits the LSTM.

#### 4 Discussion



We structure our discussion around the three research questions posed at the end of our introduction.

1. Can LSTMs extrapolate to discharge values beyond the training distribution when forced with statistically derived design precipitation events?

Our study highlights limitations in current LSTM training strategies. While LSTMs are undeniably powerful tools for modelling complex relationships in hydrological systems (Kratzert et al., 2018, 2019a; Loritz et al., 2024; Nearing et al., 2024), their response to inputs outside the training range exposes critical challenges (Acuna Espinoza et al., 2024b; Song et al., 2024). In order to use ML models responsibly, users should be aware of how the training data limit the model applicability (see also: Meyer and Pebesma, 2021).

Although we train the LSTM ensemble using state-of-the-art methods following the current benchmarks (Kratzert et al., 2019a; Lees et al., 2021; Acuna Espinoza et al., 2024b), it still underestimates discharge values with low exceedance proba-

bilities (high floods), even when these are present in the training data. For instance, although the model saw the largest flood in the training period of 183 mm d<sup>-1</sup> and 66 other events higher than the theoretical prediction limit (73 mm d<sup>-1</sup>) 20 times during training (once every epoch of training), the maximum value it could simulate is much lower (60 mm d<sup>-1</sup>). Extreme hydrological events often coincide with distinct regime shifts, such as the switch to runoff generation dominated by surface runoff, which was previously dominated by subsurface runoff. This may necessitate the model to adopt a completely different set of network weights and a unique mapping of inputs to outputs to accurately capture these phenomena. However, reallocating network capacity in this way could compromise the model's ability to simulate more common flow conditions. Thus, the model is potentially disincentivized from fitting to these rare but critical extremes effectively. Another contributing factor may be the inherent bias of minimizing the mean squared error (MSE), which disproportionately penalizes rare outliers and can lead to systematic underestimation of their magnitude. Furthermore, both the inputs and targets are frequently noisy, adding another layer of complexity to accurately capturing extreme events. While our experiments cannot definitively determine which of these factors—or their combination—is primarily responsible for the observed underestimation of extreme floods, the inherent flexibility of LSTMs suggests that this limitation is not intrinsic to the model itself. Instead, it highlights the need for an improved training strategy that better balances the representation of rare extremes and common flow conditions.







Scaling the LSTM by increasing the number of hidden states, and/or providing more training data from a broader range of hydrologic conditions, seems to be an avenue to mitigate this problem. For instance, our LSTM with 256 hidden states, trained on a combined CAMELS-US and CAMELS-CH dataset, results in improved simulations of the extreme events in our test catchments. This corroborates the intuition given by Kratzert et al. (2019a) and studied in Kratzert et al. (2024). However, the theoretical limit of the ensemble, in this case, was still well below the maximum observed training data in Switzerland and far below that of CAMELS-US. Once again, it is imprudent to state with certainty, the underlying reason or combinations thereof—whether it is the rarity of the extreme events or the training strategy which minimizes a squared error. Our study provides some indications on how we can overcome these limits: For one, our results show that stronger structural priors—as for example implemented by the hybrid-approach—can lead to behavior that is more plausible. However, we do not yet know how strong or weak the structural choices need to be (the study by Frame et al. (2022) indicates that mass conservation alone is not enough). Another potential avenue could come from the training itself: During the training process, there are no technical limits to a prediction made by the LSTM. Hence, the issue could most likely be reduced by a well-chosen training strategy. For example, changing the loss function (for instance by weighting high flow events more; Tanrikulu et al., 2024) improves the predictions for flood peaks, but is accompanied by a loss in overall performance. In this study, we tried training the LSTM with a different loss functions as well as training on more diverse datasets. Both the strategies only mitigated the issue to some extent. We believe this issue can be resolved completely and there is, indeed, a need for improvement in the way we train and setup LSTMs in hydrology. We leave the further exploration of potential solutions to future work.

2. Is the saturation of LSTM cell states the primary reason, which limits their ability to extrapolate to extreme and unprecedented hydrological conditions?

Our multi-day design precipitation experiments highlight that, saturation of the cell states can be an important reason for the threshold behavior, as increasing inputs led to large values of  $c_t$  (Eq. (A5)) for certain cells—which are then asymptotically limited to -1.1 by the tanh function. However, the theoretical limit of the LSTM derived in Kratzert et al. (2024) can only partly explain why the model does not respond to increasing inputs. The reason for this is that the other gating mechanisms can in practice saturate much earlier. Hence, one has to consider the model response as a whole and empirically, the design limit lies below the theoretical maximum from Kratzert et al. (2024). As a matter of fact, a deeper examination of the internal mechanisms—particularly the behavior of the gating functions (see Appendix A1)—showed that, most 1-day design precipitation events never reach the cell state because the input gate (Eq. (A1)) in the LSTM filters them out, or the forget gate (Eq. (A2)) discards most of the historical information. This suggests that the LSTM's inherent assumptions and structural characteristics can prevent it from effectively processing extreme inputs, leading to an underestimation of extreme high-flow events, as additional mass is effectively "deleted" (in contrast, we posit that, for low-flow events this property should not be antagonistic to the hydrological intuition, since saturation behavior naturally occurs there). In principle, an LSTM could also be built with its gating functions employing non-saturating activation functions, but this would typically introduce significant new challenges (e.g., due to vanishing gradients; Hochreiter and Schmidhuber, 1997). Non-saturating functions (e.g., Rectified Linear Units) do not naturally bound the values that flow through the network, making it harder to control the internal state dynamics. Without the built-in constraints provided by sigmoid or tanh activations, the cell states could grow without bound, potentially leading to exploding gradients and destabilized training. In this regard, it is of interest to compare the mechanism of the original LSTM with its latest iteration, the xLSTM (Beck et al., 2024) – more specifically, the sLSTM variant. It incorporates a non-saturated exponential function for the input gate. However, it also relies on additional stabilizing mechanisms that also leads to a form of saturation, ensuring that values remain within manageable ranges. In this way, while alternative architectures and activation functions might circumvent certain limitations, they often introduce new challenges related to stability and training dynamics. Ultimately, these findings again highlight that, when it comes to purely data-driven models, there is no simple, one-size-fits-all solution; rather, careful architectural choices, tailored activation functions, and potentially new inductive biases are needed to effectively capture and represent extreme events within LSTM-based models.








3. How do the inherent assumptions and structural characteristics (inductive biases) of LSTMs influence their ability to simulate realistic hydrological responses under conditions that exceed observed training ranges?

LSTMs are not just general function approximators, but are also proven to be Turing complete (Siegelmann and Sontag, 1992; Chung and Siegelmann, 2021). However, the inherent assumptions and structural characteristics of an LSTM introduce an inductive bias that can limit its ability to simulate hydrological responses when conditions strongly deviate from those observed during training. In essence, the LSTM's model structure acts as a form of prior knowledge that guides its predictions toward states that reflect its training experience (Hochreiter and Schmidhuber, 1997). The LSTM design, however, does not focus on yielding model behavior that reflects hydrological intuitions in extrapolation regimes. In case of the LSTM and the maximum runoff reaction, this is due to its reliance on saturating activation functions (which, for large precipitation values, results in an input-concave behavior) and in case of the hybrid and its use of linear reservoirs, close to linear (if the parameters

remain unchanged during the extreme event; which empirically they do, due to the saturation of the LSTM). In contrast to both models, in hydrology, we might assume a convex model behavior with increase in precipitation (assuming no changes in the other input features). Thus, we typically assume that runoff coefficients increase with increasing intensity of extreme events, as increasing area of a catchment becomes saturated (Beven et al., 2021; Kirchner, 2024). In other words, if we plotted runoff as a function of precipitation for increasingly intense events, we might observe a curve that bends upward (convex). This shape reflects the fact that once critical saturation thresholds are reached, each additional unit of rainfall generates disproportionately more runoff than before. In a single linear reservoir model, the runoff response is, for instance, inherently linear, meaning the total runoff volume (the integral of O(t) over time) remains proportional to the total rainfall input, assuming negligible losses or constraints. The runoff coefficient in such a system is constant irrespective of rainfall magnitude (approximately what we found with the hybrid model and also what we found for a single HBV model, (Seibert and Vis, 2012) locally calibrated for each of the test catchments (see Fig. B2.). In contrast, conceptual models such as TOPMODEL (Beven et al., 2021) encode clear nonlinearities due to the exponential relationship between subsurface flow and water-table depth. This nonlinearity implies a substantial increase in runoff generation as saturation thresholds within the catchment are approached, resulting in runoff coefficients that vary strongly with antecedent moisture conditions and rainfall magnitude. It is thereby obvious that this is a simplified perspective and that runoff generation across catchments, depends on multiple factors such as topography, soil characteristics, land use, and antecedent moisture conditions. For instance, Froidevaux et al. (2015) showed in a study conducted in 100 Swiss catchments that 0 to 3 days of accumulated precipitation is the main driver of floods, while longer-term (4 days to 1 month) antecedent precipitation and hydrological conditions have only weak, region-specific effects and are negligible in Alpine catchments. While Staudinger et al. (2025) found that only 18-44% of extreme annual floods coincided with maximum precipitation, highlighting the crucial role of antecedent soil moisture and snow storage. The sensitivity of flood peaks to an increase in maximum precipitation varies significantly, however, at a fundamental level, one would generally expect runoff coefficients to increase or at least remain the same with increasing rainfall, particularly under extreme precipitation scenarios. But interestingly, our analysis revealed that the LSTM model exhibited an unexpected and physically counterintuitive trend: runoff coefficients start decreasing with increasing precipitation magnitudes, especially for extreme precipitation values. This is particularly true for catchments with higher runoff generation. If we trust our hydrological theory, this knowledge should also be reflected in the "inductive bias" of the model we are using. In reality, hydrology is much more complex, and we could observe concave hydrological responses to increasing precipitation, but the a-priori assumption of a convex reaction seems reasonable.







The hybrid model (and the HBV model (Appendix B)) effectively avoids the unrealistic behavior observed in the standalone LSTM by enforcing an almost linear behavior due to its use of linear reservoirs. Under the design precipitation events the LSTM component within the hybrid model does saturate, showing a similar behavior as the purely data driven approach. This implies a theoretical prediction limit to every parameter of the subsequent HBV models, which is the upper limit of its parameter range specified during the initialization. However, similar to a stand-alone LSTM, the LSTM component of the hybrid model does not reach full saturation for any of the observed extreme events, and the saturated parameters of the HBV

component still remain well below their theoretical prediction limits. Crucially, the conceptual structure of the hybrid model ensures that predicted discharges increase consistently with increasing precipitation. This alignment with hydrological principles allows the hybrid model to provide predictions that remain hydrologically plausible even when the model is forced with inputs outside the observed regime. In other words, the structural choices of the hybrid-model effectively mitigate the saturation behavior observed in the stand-alone LSTM—making the hybrid approach more suitable for applications like infrastructure design where plausible extrapolation behavior is essential. Asserting whether the actual behavior reflects a real-world response of the underlying basin, and whether it is actually meaningful to use these models in such a way, is beyond the scope of this study.

For operational flood forecasting, the situation may differ. Recent work by Nearing et al. (2024) highlights the potential advantages of LSTMs over classical hydrological models, particularly when trained on a global database. Our results support this, showing that in catchments with low runoff generation, the LSTM behaves in a hydrologically consistent manner. Additionally, the stand-alone LSTM offers numerous advantages over classical hydrological models. For instance, its flexible use of embedding layers enables the model to seamlessly transition between different temporal frequencies and switch between simulation and forecasting modes (Acuña Espinoza et al., 2024). This adaptability makes LSTMs a powerful tool in operational settings, where diverse conditions and forecasting needs must be addressed efficiently. By emphasizing on high-flow events (Tanrikulu et al., 2024) during training or employing data augmentation techniques like weather generators combined with classical hydrological models (Nguyen et al., 2021), the simulation of extreme events included in the training data could probably be improved.

#### 5 Conclusion


This study investigates the ability of LSTMs to extrapolate under extreme rainfall—runoff conditions and compares their performance with a hybrid model. Based on our findings, we conclude the following:

- Limitations of LSTMs: State-of-the-art LSTMs struggle to predict discharge values beyond a theoretical prediction limit, and this limit is below the range of the training data.
  - Saturation of LSTM states: While saturation of LSTM cell states contributes to limiting the model's ability to simulate extreme hydrological events, the gating mechanisms play a significant role in filtering or discarding information, especially during 1-day design precipitation events.
- Inconsistent runoff responses: Increasing (extreme) intensity of design precipitation events leads to decreasing runoff
  coefficients, contrary to the hydrological expectation. This highlights structural limitations in the LSTM architecture for
  hydrological extreme value simulation.

Hybrid model benchmark: The hybrid model aligns better with hydrological principles, demonstrating consistent scaling
of discharge with increasing extreme precipitation. Its mass-conserving structure and use of conceptual hydrological
components make it more robust under extreme forcing conditions.


- Potential for improvement: Increasing the number of LSTM hidden states and training on larger, more diverse datasets can raise the theoretical and design prediction limits. However, these adjustments do not fully address the observed limitations, particularly during the 1-day events. Incorporating stronger structural priors, or adapting training strategies which weigh extreme events more during optimization, could mitigate these issues.
- Every modeling approach has inherent limitations within its scope of application. While the constraints of conceptual hydrological models are well understood, the same cannot be said for deep learning models, where such limitations remain less explored. We argue that addressing these gaps is crucial for advancing their utility in hydrological applications. The limitations outlined above are not beyond resolution; they represent opportunities for further development. Future research should focus on refining LSTM architectures to better align with hydrological principles, improving training strategies to give greater weight to extreme events during optimization, and exploring innovative hybrid approaches that combine the strengths of data-driven and process-based models. By addressing these challenges, we can move closer to unlocking the full potential of deep learning in hydrological modelling, particularly under extreme forcing conditions. All of the above stated limitations can potentially be overcome, and we believe that future research should focus on refining LSTM architectures, improving training strategies, and exploring and optimizing new hybrid approaches.
- Code availability. All the codes for model training, testing, design experiments and plotting the results presented in this paper are available at https://doi.org/10.5281/zenodo.14771377. This also contains the CAMELS-CH and the CAMELS-US dataset for the ease of reproduction of results.

Data availability. The CAMELS-US dataset is freely available at https://doi.org/10.5065/D6MW2F4D (Newman et al., 2015; Addor et al., 2017). The CAMELS-CH dataset is freely available at https://doi.org/10.5281/zenodo.7784632 (Höge et al., 2023). Extreme value analyses for Switzerland is available at https://www.meteoswiss.admin.ch/services-and-publications/applications/standard-period.html (MeteoSwiss, 2022)

# **Appendix A: Model Inputs and LSTM Equations**

# A1 Equations describing the LSTM

The LSTM forward pass can be mathematically represented by the following:

$$i_t = \sigma(W_i x_t + U_i h_{t-1} + b_i) \tag{A1}$$

$$f_t = \sigma \left( W_f x_t + U_f h_{t-1} + b_f \right) \tag{A2}$$

$$g_t = \tanh\left(W_q x_t + U_q h_{t-1} + b_q\right) \tag{A3}$$

$$o_t = \sigma(W_o x_t + U_o h_{t-1} + b_o) \tag{A4}$$

$$c_t = f_t \odot c_{t-1} + i_t \odot g_t \tag{A5}$$

$$\mathbf{495} \quad \boldsymbol{h_t} = \boldsymbol{o_t} \odot \tanh(\boldsymbol{c_t}) \tag{A6}$$

where  $i_t$ ,  $f_t$ , and  $o_t$  are the input gate, forget gate, and output gate, respectively,  $g_t$  is the cell input and  $x_t$  is the network input at time step t, and  $h_{t-1}$  is the recurrent input,  $c_{t-1}$  the cell state from the previous time step. W, U, and b are learnable parameters for each gate, where subscripts indicate which gate the particular weight matrix/vector is used for,  $\sigma$  is the sigmoid function, tanh is the hyperbolic tangent function, and  $\odot$  is element-wise multiplication.

# 500 A2 List of the CAMELS-CH and CAMELS-US forcing variables and catchment attributes used for training

Table A1 gives the description of the static and dynamic inputs to the LSTM and hybrid models.

Table A1: Dynamic and static inputs used to train the <sup>1</sup>LSTM ensembles using the CAMELS-CH dataset, <sup>2</sup>LSTM ensembles using CAMELS-CH and CAMELS-US dataset combined and <sup>3</sup>hybrid model ensembles <sup>4</sup>explicit input to the HBV models in the hybrid model

| CAMELS-CH                    | CAMELS-US                  | Description                                                  |
|------------------------------|----------------------------|--------------------------------------------------------------|
| Dynamic Inputs               |                            |                                                              |
| precipitation (mm $d^{-1}$ ) | prcp (mm d <sup>-1</sup> ) | Observed daily summed precipitation <sup>1,2,3</sup>         |
| temperature_min (°C)         | tmin (°C)                  | Observed daily minimum temperature <sup>1,2,3</sup>          |
| temperature_max (°C)         | tmax (°C)                  | Observed daily maximum temperature <sup>1,2,3</sup>          |
| rel_sun_dur (%)              |                            | Observed daily averaged relative sunshine (solar irradiance  |
|                              |                            | $\geq$ 200 W m-2) duration <sup>1,3</sup>                    |
| swe (mm)                     |                            | Observed daily averaged snow water equivalent <sup>1,3</sup> |
| pet_sim (mm $d^{-1}$ )       |                            | Simulated daily averaged potential evapotranspira-           |
|                              |                            | tion (Penman-Monteith equation without interception          |
|                              |                            | $correction)^{3,4}$                                          |

| CAMELS-CH                            | CAMELS-US                            | Description                                                   |
|--------------------------------------|--------------------------------------|---------------------------------------------------------------|
| Static Inputs                        |                                      |                                                               |
| area (m <sup>2</sup> )               | area_gages2 (km <sup>2</sup> )       | catchment area                                                |
| elev_mean (m a.s.l.)                 | elev_mean (m a.s.l.)                 | Mean elevation within catchment                               |
| slope_mean (°)                       | slope_mean (m $km^{-1}$ )            | Catchment mean slope over all grid cells                      |
| sand_perc (%)                        | sand_frac (%)                        | Percentage sand                                               |
| silt_perc (%)                        | silt_frac (%)                        | Percentage silt                                               |
| clay_perc (%)                        | clay_frac (%)                        | Percentage clay                                               |
| porosity (-)                         | soil_porosity (-)                    | Volumetric porosity                                           |
| conductivity (cm $h^{-1}$ )          | soil_conductivity (cm $h^{-1}$ )     | Saturated hydraulic conductivity                              |
| glac_area (km²)                      |                                      | Glacier area of Swiss glaciers per catchment                  |
| dwood_perc (%)                       |                                      | Percentage of deciduous forest                                |
| ewood_perc (%)                       |                                      | Percentage of coniferous forest (evergreen)                   |
| crop_perc (%)                        |                                      | Percentage of agriculture                                     |
| urban_perc (%)                       |                                      | Percentage of urban and settlements                           |
| reservoir_cap (ML)                   |                                      | Total storage capacity of reservoirs in megaliters            |
| $p_mean (mm d^{-1})$                 | $p_mean (mm d^{-1})$                 | Mean daily precipitation                                      |
| pet_mean (mm $d^{-1}$ )              | pet_mean (mm $d^{-1}$ )              | Mean daily potential evapotranspiration (PET; Pen-            |
|                                      |                                      | man-Monteith equation without interception correction)        |
| p_seasonality (-)                    | p_seasonality (-)                    | Seasonality and timing of precipitation (estimated using      |
|                                      |                                      | sine curves to represent the annual temperature and precip-   |
|                                      |                                      | itation cycles, positive (negative) values indicate that pre- |
|                                      |                                      | cipitation peaks in summer (winter), and values close to      |
|                                      |                                      | zero indicate uniform precipitation throughout the year).     |
|                                      |                                      | See Eq. (14) in Woods (2009))                                 |
| frac_snow (-)                        | frac_snow (-)                        | Fraction of precipitation falling as snow, i.e., while tem-   |
|                                      |                                      | perature is < 0 °C                                            |
| high_prec_freq (d yr <sup>-1</sup> ) | high_prec_freq (d yr <sup>-1</sup> ) | Frequency of high-precipitation days (≥ 5 times mean          |
|                                      |                                      | daily precipitation)                                          |
| low_prec_freq (d yr <sup>-1</sup> )  | low_prec_freq (d yr <sup>-1</sup> )  | Frequency of dry days ( $

**Figure B1.** Model performance comparison in terms of cumulative distribution function (CDF) of Nash-Sutcliffe Efficiency (NSE) for PREVAH, conceptual model, LSTM (ensemble mean) and hybrid model (ensemble mean) for (a) 196 CAMELS-CH catchments and (b) subset of 25 catchments identified for design experiments

# **B1** Conceptual Model Description and Performance



To enable model comparison across the entire range of models, in addition to the LSTM and hybrid model ensembles, we locally trained stand-alone conceptual models for individual catchments. The conceptual model is a variant of the HBV model (Seibert, 2005) plus a unit hydrograph (UH) routing, with a total of 14 parameters (12 HBV and 2 UH routing parameters). For brevity, we refer the reader to Seibert (2005) for a detailed description of the HBV model. The models are calibrated locally for every catchment using the "differential evolution adaptive metropolis" (DREAM) (Vrugt, 2016) algorithm, which is implemented within the SPOTPY (Statistical Parameter Optimization Tool for Python) library (Houska et al., 2015), as done in the CAMELS-DE dataset (Loritz et al., 2024). Using the best catchment-specific calibration parameters, the models were tested for the experimental set-up described in Section 2.5. The calibration period and evaluation periods for the conceptual models is the same as the training and testing periods mentioned in Table 1. Fig. B1 panel (a) presents the CDF of the NSE for 196 catchments from the CAMELS-CH identified in Section 2.1 and panel (b) shows the performance of the models for the subset of 25 catchments identified for the design experiments. Though the HBV model (median NSE 0.64) outperforms the PREVAH model (median NSE 0.50) for overall performance, the HBV model fails to accurately simulate runoff during winter periods for some catchments, potentially owing to it's rather simple temperature degree snow module.

**Figure B2.** Model simulation comparison for 25 catchment specific 1-day events with the highest runoff generation. Variation within the LSTM and Hybid model ensembles is represented by the whiskers on their respective plots. The HBV results are from a single model. As the LSTM prediction approaches the theoretical prediction limit, saturation behavior is most pronounced.

# B2 Model comparison for design events simulation for 25 catchment-specific events

A comparison of the simulated discharge from the three models for 25 catchment specific 1- and 3-day events is given in, Figs. B2 and B3 respectively. The events shown in these figures are those, for which the LSTM has the highest runoff response. For such events, the LSTM is most likely to exhibit the saturation behavior as it nears its prediction limits. For the 1-day events (see Fig. B2), the saturation behavior in the LSTM is more apparent for events with runoff generation closer to the "design limit" (see (a1), (a2), (b3), (b5), (c2), (d1), (d2), (d5), (e1), (e5) in Fig. B3). For most of the events, the response of the conceptual model is smaller than the LSTM, but it shows greater increase with increasing intensity of precipitation. For the 3-day events, owing to less intense daily precipitation value, the saturation behavior of the LSTM is observed only for a few events (see panels (a2), (b3), (c4), (d3) and (e2) in Fig. B3). The discrepancy between the hybrid and the LSTM simulations is much smaller for these events as compared to the 1-day events. For most of the events, the conceptual and the hybrid model responses are almost comparable.

# Appendix C: Effect of increased network size and larger training datasets on theoretical prediction limit and design limits

As mentioned in Section 3.2 of this paper, increasing the number of hidden states, and/or training the LSTMs on larger datasets, increases the theoretical prediction limit as given in Table C1. LSTMs with more hidden states and/or trained on larger dataset also simulate higher runoff for the design precipitation values. Nevertheless, this response, too, is concave (Fig. C1), unlike the hybrid model response.

**Table C1.** Theoretical prediction limits and design limits from design experiments for different LSTM networks.  $max(y_{obs})$  indicates the maximum observed target value during the training period from 01.10.1995 to 30.09.2005.

\*results from this model presented in Section 3 of the main text


| LSTM Network | Number | of | Training Dataset             | $\max(y_{obs})$          | Theoretical             | Design Limit          |
|--------------|--------|----|------------------------------|--------------------------|-------------------------|-----------------------|
|              | Nodes  |    |                              |                          | Prediction              |                       |
|              |        |    |                              |                          | Limit                   |                       |
|              |        |    |                              | ${\rm mm}\;{\rm d}^{-1}$ | ${\rm mm}~{\rm d}^{-1}$ | ${ m mm}~{ m d}^{-1}$ |
| LSTM_CH*     | 64     |    | 229 CAMELS-CH catchments     | 183                      | 73                      | 60                    |
| LSTM_CH      | 256    |    | 229 CAMELS-CH Catchinents    | 103                      | 120                     | 76                    |
| LSTM_US_CH   | 64     |    | 229 CAMELS-CH                | 299                      | 115                     | 84                    |
| LSTM_US_CH   | 256    |    | and 531 CAMELS-US catchments | <b>∠</b> ∃∃              | 193                     | 110                   |

**Figure B3.** Model simulation comparison for 25 catchment specific 3-day events with the highest runoff generation. Variation within the LSTM and Hybid model ensembles is represented by the whiskers on their respective plots. The HBV results are from a single model.

Figure C1. Additional LSTM networks' and hybrid model ensemble simulation for 3 catchment specific events.

# Appendix D: Effect of training an LSTM with modified loss function and modified activation functions on design limits

In order to investigate training strategies that overcome the characteristic behavior of the LSTM, we trained an LSTM with a modified loss function instead of the basin-averaged NSE suggested by Kratzert et al. (2019a). The modified loss function, in this case, weighs the maximum of the squared errors between the observation and the simulation by a factor, thus forcing the LSTM to simulate tail end values of the discharge distribution better. In other attempts, we focused on replacing the *tanh* activation function in Eq. (A6) to overcome the saturation in the LSTM. Replacing it with a non-saturating *softplus* activation function, made the LSTM training unstable, thwarting our efforts in this direction. We then implemented the sLSTM variant of the xLSTM (Beck et al., 2024), as it replaces the *sigmoid* activation in Eqs. (A1) and (A2) with an *exponential* activation function. Such a replacement is hypothesized to enable better transmission of the extreme input signal through the input and the forget gates of the sLSTM. In Appendix D1, we first describe the modified loss function (MSE<sup>+</sup>) and the mathematical equations describing the forward pass of the sLSTM. We also give a brief description of the training and testing methods for these models. In Appendix D2 we present the results from these models for the same events shown in Fig. 2.

# D1 Methods Description: LSTM $_{MSE^+}$ and sLSTM forward pass





An ensemble of 5 LSTM networks was trained with a modified loss function given in Eq. (D1), henceforth referred to as the LSTM $_{\rm MSE^+}$ . Another ensemble of 5 sLSTM networks was trained and the equations describing the forward pass of the sLSTM are described in Eqs. (D2) to (D11). The hyperparameters and the training and testing data split for both the ensembles were the same as mentioned in Table 1. Thus, the LSTM $_{\rm MSE^+}$  differs from the stand-alone LSTM only in terms of the loss function and the sLSTM differs only in its forward pass.

$$MSE^{+} = MSE + k \cdot \max((obs - sim)^{2})$$
(D1)

where  $MSE^+$  is the modified loss function, k is a factor (= 0.2 in this study), obs and sim are the observed and simulated discharge time series respectively.

$$i_t = \exp\left(W_i x_t + U_i h_{t-1} + b_i\right)$$
(D2) 
$$i'_t = \exp\left(\log\left(i_t\right) - m_t\right)$$

$$f_t = \exp(W_f x_t + U_f h_{t-1} + b_f)$$
 (D3)  $f'_t = \exp(\log(f_t) + m_{t-1} - m_t)$ 

$$o_t = \sigma(W_o x_t + U_o h_{t-1} + b_o)$$

$$(D4) \qquad c_t = f_t' c_{t-1} + i_t' z_t$$

$$(D9)$$

$$z_t = \tanh(W_z x_t + U_z h_{t-1} + b_z)$$
 (D5)  $n_t = f_t' n_{t-1} + i_t'$ 

$$m_t = \max(\log(f_t) + m_{t-1}, \log(i_t))$$
 (D6)  $h_t = o_t\left(\frac{c_t}{n_t}\right)$ 

where  $i_t$ ,  $f_t$ , and  $o_t$  are the input gate, forget gate, and output gate, respectively,  $z_t$  is the cell input,  $x_t$  is the network input at time step t,  $h_t$  is the recurrent input,  $c_t$  the cell state,  $n_t$  is the normalizer state,  $m_t$  is the stabilizer state and  $i_t'$  and  $f_t'$  are the stabilized input and forget gates respectively. W, U, and b are learnable parameters for each gate, where subscripts indicate which gate the particular weight matrix/vector is used for,  $\sigma$  is the sigmoid function, tanh is the hyperbolic tangent function, and exp is the exponential function. The sLSTM architecture replaces the sigmoid activation function in the input and the forget gates with the exponential activation and in order to prevent overflow, a stabilizer state  $m_t$  is introduced to stabilize these gates.

# D2 Design experiments results: $LSTM_{MSE+}$ and sLSTM forward pass

Table D1. Predictions for design events (and Theoretical Prediction Limits) for LSTM, LSTM $_{\mathrm{MSE}^{+}}$  and sLSTM for three most runoff reactive design events

| Gauge ID | Catchment         | Prediction for 1-day design experiment at ARI 300-year (mm $d^{-1}$ ) |                           |                          |  |  |  |
|----------|-------------------|-----------------------------------------------------------------------|---------------------------|--------------------------|--|--|--|
|          |                   | Original LSTM from                                                    | $\rm LSTM_{MSE^+}$        | sLSTM                    |  |  |  |
|          |                   | this study                                                            |                           |                          |  |  |  |
|          |                   | $(73 \text{ mm d}^{-1})$                                              | $(110 \text{ mm d}^{-1})$ | $(66 \text{ mm d}^{-1})$ |  |  |  |
| 2087     | Andermatt         | 48                                                                    | 53                        | 40                       |  |  |  |
| 2494     | Pollegio-Campagna | 43                                                                    | 44                        | 36                       |  |  |  |
| 2461     | Magliaso-Ponte    | 60                                                                    | 63                        | 52                       |  |  |  |

Results from the two models for the same events shown in Fig. 2 are summarized in table Table D1. The LSTM<sub>MSE+</sub> ensemble has an improved mean theoretical prediction limit of about 101 mm d<sup>-1</sup>, but the design limits did not show a corresponding improvement. Such an ensemble also had a lower median performance (median ensemble NSE 0.75) for overall runoff simulation. The sLSTM ensemble on the other hand had a slightly better overall performance (media ensemble NSE 0.78) as

compared to the LSTM<sub>MSE+</sub>, but did not the match the performance of the LSTM. The design experiments with the sLSTM ensemble show a decreased theoretical prediction limit of about 66 mm d<sup>-1</sup>. This is accompanied by a decrease in the design limits as well. These results warrant efforts to further explore more such training strategies and network architectures.

Author contributions. The idea for the paper was proposed by RL. Codes developed by EAE were used for training the models. Model training and testing, the design experiments and analysis were done by SB, and results were discussed with RL. The draft was prepared by SB and reviewed and edited by all authors. Funding was aquired by RL. All authors have read and agreed to the current version of the manuscript.

Competing interests. Some authors are members of the editorial board of HESS.

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
