# Peer review of "Unveiling the Limits of Deep Learning Models in Hydrological Extrapolation Tasks"

_EGUsphere, 2025_

## Referee Comment (RC1)

**"Unveiling the Limits of Deep Learning Models in Hydrological Extrapolation Tasks"**

**Manuscript #*2025-425***

B. Kraft

March 14, 2025

**Short summary and highlights**

This study evaluates how well LSTMs, both in a standard setup and as part of a hybrid LSTM/process-based model, generalize to extreme conditions. The authors train their models on a subset of catchments from the CAMELS-CH dataset. Some experiments also incorporate CAMELS-US data. They then test the models using extreme precipitation events derived from extreme value analysis. These events cover 1- to 5-day precipitation totals with return intervals ranging from 1 to 300 years. On the hold-out set—a time period excluded from training—the LSTM significantly outperforms the hybrid model. However, the main focus is on how the models respond to extreme precipitation inputs.

The experiments reveal a strong saturation effect in the LSTM when exposed to large precipitation events. In contrast, the hybrid model does not exhibit this saturation and maintains a quasi-linear response. It predicts higher runoff values for larger precipitation events. While the LSTM's saturation effect has been observed and discussed in prior work, it is surprising that it fails to predict output values even within the range seen during training. In practice, this saturation occurs at levels lower than the theoretical maximum.

The study highlights a major limitation of the widely used LSTM architecture in hydrological applications. The findings raise concerns about whether LSTMs are suitable for extreme event prediction.

**Major remarks**

1. I see potential for making the study much more impactful by testing solutions for the saturation effect. The authors speculate a lot in the discussion, and I was a bit disappointed that these ideas were not tested. However, I respect the authors' decision to not include further analysis. I do not insist on further experiments, but I would find them very interesting.

2. The hybrid model receives 730 days of input, while the LSTM only receives 365 days. The authors argue that the hybrid model needs the additional data for spin-up of the internal states. This makes sense, but the LSTM also needs to initialize its states. If the LSTM is used in a many-to-many setup (where the prediction period corresponds to the input period), the warm-up period should be the same for both models.

3. Why does the hybrid model receive an additional input? I believe the comparison between the models would be fairer if both models received exactly the same input data. Please clarify.

4. The synthetic precipitation data is extracted for a selection of catchments within a distance of 2.5 km. If I understand correctly, the model is fed with point observations, while it was trained on the average precipitation over the catchment area. This could be a source of error. Please discuss or clarify.

5. I would appreciate a brief analysis of the extreme values within the test data. How do the predictions compare to the observations? If you see the same saturation in the test data, this would strengthen your argument. It would also rule out that the experimental setup (changing precipitation but not temperature, etc.) is the cause of the saturation. You mention this in the discussion (L365-370), and I suggest showing it instead of speculating.

**Minor remarks**

Here I list some typos and suggestions for improving clarity:

L6 In the first reading, I did not understand where the 73 mm d$^{-1}$ came from. I suggest clarifying this, e.g., "...show that this limit (which we have calculated for this study to be 73 mm d$^{-1}$) is below ...".

L68 "plausible" instead of "possible"?

Sect. 2.1&2.2 The description of the data splitting and how it is used to train the model (L104-107 and L116-117) is a bit misplaced. I suggest moving it to the Methods section.

L199 Could it be that this is because you use different catchments in your study?

Fig. 2a-c Consider adding the prediction limit also to the top row, also in Fig. 3a-b.

Fig. 3 The second panel should be (b), not (a).

L318 "...can lead to  behavior that is more plausible ....."

L323 "Alternatively, one can also think to directly train for the warranted behavior." This is a bit vague. What do you mean exactly?

L351 Since you already speculate: Would using other NN architectures be a solution?

L363 I suggest replacing "ceteris paribus" with an English term.

L377 The hybrid model coefficients (the outputs of the LSTM) can, in theory, also saturate. Consider mentioning this, or, even better, calculating the theoretical limits.

L380 Consider making a sentence out of the parentheses. This would make the sentence easier to read.

Tab. A1 Is mean temperature missing in the table, or did you not use it?

Tab. C1 Could you add the actual prediction limit (the "design limit") to the table?

---

## Author Comment (AC3)

**Reply to RC2: 'Comment on egusphere-2025-425', Anonymous**

We thank the referee for the detailed evaluation of our manuscript and the insightful comments, which will help us improve the paper. In this document we address the comments, questions and suggestions posted by the referee. Please find the referee remarks in blue and our response in black.

General comment

This paper compares a stand-alone LSTM model with a hybrid HBV-LSTM model on the CAMEL-CH dataset. It also examines the impact of training the stand-alone LSTM on the CAMEL-US dataset, alongside CAMEL-CH, and also using 256 nodes instead of 64. The main focus of the discussion is on the ability of both models (stand-alone LSTM and hybrid) to show a linear pattern between simulated peak flows and rainfall when applying "synthetic" rainfall far higher the observed ones. The results clearly show the impossibility for the stand-alone LSTM model to exibit this linear pattern due to simulated discharges tending to a limit values as predicted by a previous study. But more interestingly, it clearly shows that this observed limit is far lower than the theoretical limit expected by the authors.

In my opinion, this paper is very interesting as it is the first to clearly and honestly address the limitations of LSTM models in hydrology.

I recommend publication after revision.

Response: We thank the reviewer for this interesting and positive assessment of our work.

Main comments

My main comments are:

1) Even though it is outside the scope of the paper, I would have appreciated a "deeper" and "fairer" comparison between the stand-alone LSTM model and the hybrid HBV-LSTM model. The paper is short (only 3 figures of results in the main text), there is room for that. My main criticism

is that no hyperparameter (HP) tuning is done for either model. The HP values are simply taken from previous studies. I think that the results could be different if a proper fine tuning was done for each model.

Response: We agree with the reviewer that hyperparameter tuning is an important part of any deep-learning model training. However, for our setting, the hyperparameters are not as critical as for benchmarking settings. Hence, we used the ones that have been identified in previous studies (Acuna Espinoza et al., 2024 and Kratzert et al., 2019). In some sense, this might be suboptimal. Nevertheless, we tried several hyperparameter combinations. In that regard, we tested the LSTM for a various number of hidden states, ranging from 8 to 2048 and obtained the same results. As the primary goal of the study is to highlight the saturation behavior in the LSTM, the nature of our results will not change even with a better set of hyperparameters. Having said that, we will implement a hyperparameter tuning for the stand-alone LSTM. If it changes the nature of our results significantly, we shall revise the manuscript accordingly. However, we believe that it is justified to adopt the same hyperparameters from the LSTM for the Hybrid model, for a fair comparison. Moreover, as it wouldn't change the nature of our results by large, hyperparameter tuning for the hybrid model might not be worth the computational demand for this study.

2) For analysis, it would also be very interesting to see the results for a single HBV model as a benchmark, which is very "cheap" to calibrate locally. Is there an improvement and is it "worth" the huge amount of data and GPU time required to process it? For example, the authors added US CAMEL data to their CH CAMEL learning dataset and moved from 64 to 256 nodes, which would have required a considerable amount of additional resources, but they don't show the corresponding improvement.

Response: We would first like to address the reviewer's comments regarding increased computational demand for the different LSTM models in our study. Technically, with four times the original number of hidden states (256 instead of 64), the increased computational demand is 16 times the original. In terms of time required: training a single LSTM using CAMELS-CH takes about ~900s (64 hidden states) and ~1700s (256 hidden states) on a GPU type V100. Including

CAMELS-US in the training data requires increased time of ~6000s and ~13500s respectively. This is not necessarily a concern for us, since we specifically wanted to test the effect of increased LSTM size and more training data. The fact that the models don't show corresponding improvement was only known as a result of these experiments. We agree that it will be interesting to see the results from locally calibrated conceptual models for our design experiments. We will implement a single HBV model calibrated locally for catchments in our study. The manuscript will be revised accordingly to include these results, either in the main text or in an Appendix.

3) As the paper focuses on extremes, I also think that the evaluation against the observed runoff should not be limited to the NSE criteria as in Fig. 1 (which is the only figure presenting models performances), but should include a deeper analysis, including for example signatures calculated on flood events.

Response: Thank you for your suggestion. It is a good idea, and we will consider including additional metrics in the revised manuscript.

4) The same comment applies to the second part of results (Figs. 2 and 3, using synthetic rainfall): only 1 flood for 3 catchments (a little more in the appendix), whereas the authors have thousands of examples. A synthetic metric should be found that "summarises" the different observed behaviours (between catchments, but also for the same catchment but under different conditions). A "visual" analysis on a few examples, as in this paper, is a first step to draw first hypotheses. But then these hypotheses should be tested in depth.

Response: We only focus on the three flood events shown in Figures 2 and 3 because these events highlight the saturation behavior of the LSTM most prominently. We summarize the overall trend in our results in lines 249 to 251. From the reviewer's comment, we assume that this is not discussed clearly enough, and we shall rephrase this for better clarity in the next revision. Regarding the reviewer's suggestion of developing a synthetic metric: we believe that at this stage, developing such a metric is best left as a part of future work, since it can be an intensive task. We will however address this discussion in the revised manuscript and speak to the potential of developing such a metric, as suggested here by the reviewer.

5) This last point (the need for a synthetic metrics that allows a "deep" analysis) leads me to my main comment. The authors don't clearly explain why, from a hydrological point of view, peak discharge should increase linearly with extreme rainfall. I fully agree with this, and even if it seems obvious, I think it would be valuable to anchor the paper with more basic hydrological references. In terms of synthetic metrics, I would, for example, calculate a regression coefficient between peak discharge and synthetic rainfall and see how it changes as a function of rainfall, as in the paper, but also as a function of the initial moisture content before a flood and/or the runoff coefficient. I would also not look at flood by flood, but try to find a graphical representation of all floods and catchments together.

Response: The sensitivity of flood peaks to an increase in maximum precipitation likely varies significantly across catchments, depending on multiple factors such as topography, soil characteristics, land use, and antecedent moisture conditions (as correctly highlighted by the reviewer). Given these complexities, we believe it is challenging to identify a consistent, clear signal across a large-sample dataset covering Switzerland, with its diverse hydrological regimes. Nonetheless, at a fundamental level, one would generally expect runoff to increase with increasing rainfall, particularly under extreme precipitation scenarios. For instance, in a simple linear reservoir model, the runoff response is inherently linear, meaning the total runoff volume (the integral of Q(t) over time) remains proportional to the total rainfall input, assuming negligible losses or constraints. Thus, the runoff coefficient in such a system is constant irrespective of rainfall magnitude. In contrast, conceptual models such as the TOPMODEL (Beven et al., 2021) demonstrate clear nonlinearities due to the exponential relationship between subsurface flow and water-table depth. This nonlinearity implies a substantial increase in runoff generation as saturation thresholds within the catchment are approached, resulting in runoff coefficients that vary with antecedent moisture conditions and rainfall magnitude. Interestingly, our analysis revealed that the LSTM model exhibited an unexpected and physically counterintuitive trend: runoff coefficients start decreasing with increasing precipitation magnitudes, especially for extreme precipitation values. This is particularly true for catchments with higher runoff generation. Motivated by the reviewer's suggestion, we will explore graphical representations of this phenomenon and include them if a consistent spatial pattern emerges

across Switzerland. Additionally, we will refine the manuscript by clearly articulating, from a hydrological perspective, what physically reasonable runoff responses should look like, and explicitly discuss the limitations observed in the LSTM predictions for extreme events.

Minor comments

L100 : why did not you do a hyperparameter tuning? (a LSTM expert told me one day that hyperparameter training is absolutely required in any case, and that, if "hydrologists" don't have the necessary GPU resource, they should not use LSTM)

Response: Kindly refer to our response to the reviewer's first major comment. While hyperparameter tuning can improve the overall model performance, the behavior of the LSTM under our test conditions comes from its inductive bias and will hence not change fundamentally.

L200: You should give more details on the models performances, for instance using flood signatures

Response: Additionally to the answer above we would also like to highlight that such research has been presented extensively in the studies by Acuna Espinoza et al. (2024) and Frame et al. (2022). The hybrid model is acting mainly as a benchmark here.

L224: The results for the 256 node LSTM and/or the training using US-CAMEL should be presented in Fig .1 and discussed. Does this huge amount of additional data improve models performances?

Response: We believe it is better for a reader's comprehension to only include results from the stand-alone LSTM with 64 hidden states trained on the CAMELS-CH and the hybrid model, in Figure 1.

L235: You should do more clearly the link with basic hydrological processes, such as soil saturation and the effect of initial humidity condition.

Response: We will add a stronger hydrological perspective to the discussion in the revised manuscript.

Figure 2 and 3 : the terme "observation" is misleading. There is no observed discharges in this figure.

Response: Thank you for highlighting this disparity. We will change the legend for better clarity in the next revision.

L260: this affirmation is supported only by 1 flood over 3 catchments. You should try to exhibit that using much more discharge simulation (...that you have)

Response: We will consider including more results from the 3-day and 5-day events, either in the main text or as an appendix.

L299 : "Extreme hydrological events often coincide with distinct regime shifts": I fully agree but could you explain what do you mean to a "non-hydrogist", in term of involved processes.

Response: We will rephrase this in the revised manuscript.

References:

Acuna Espinoza, Eduardo, Ralf Loritz, Frederik Kratzert, Daniel Klotz, Martin Gauch, Manuel Álvarez Chaves, Nicole Bäuerle, and Uwe Ehret. 2024. "Analyzing the Generalization Capabilities of Hybrid Hydrological Models for Extrapolation to Extreme Events." https://doi.org/10.5194/egusphere-2024-2147.

Beven, Keith J., Mike J. Kirkby, Jim E. Freer, and Rob Lamb. 2021. "A History of TOPMODEL." *Hydrology and Earth System Sciences* 25 (2): 527–49. https://doi.org/10.5194/hess-25-527-2021.

Frame, Jonathan M., Frederik Kratzert, Daniel Klotz, Martin Gauch, Guy Shalev, Oren Gilon, Logan M. Qualls, Hoshin V. Gupta, and Grey S. Nearing. 2022. "Deep Learning Rainfall–Runoff Predictions of Extreme Events." *Hydrology and Earth System Sciences* 26 (13): 3377–92. https://doi.org/10.5194/hess-26-3377-2022.

Feng, D., Liu, J., Lawson, K., and Shen, C.: Differentiable, Learnable, Regionalized Process-Based Models With Multiphysical Outputs can Approach State-Of-The-Art Hydrologic

Prediction Accuracy, Water Resources Research, 58, e2022WR032 404, https://doi.org/https://doi.org/10.1029/2022WR032404, e2022WR032404 2022WR032404, 2022.

Houska, Tobias, Philipp Kraft, Alejandro Chamorro-Chavez, and Lutz Breuer. 2015. "SPOTting Model Parameters Using a Ready-Made Python Package." Edited by Dafeng Hui. *PLOS ONE* 10 (12): e0145180. https://doi.org/10.1371/journal.pone.0145180.

Kratzert, Frederik, Daniel Klotz, Guy Shalev, Günter Klambauer, Sepp Hochreiter, and Grey Nearing. 2019. "Towards Learning Universal, Regional, and Local Hydrological Behaviors via Machine Learning Applied to Large-Sample Datasets." *Hydrology and Earth System Sciences* 23 (12): 5089–5110. https://doi.org/10.5194/hess-23-5089-2019.

Vrugt, Jasper A. 2016. "Markov Chain Monte Carlo Simulation Using the DREAM Software Package: Theory, Concepts, and MATLAB Implementation." *Environmental Modelling & Software* 75 (January):273–316. https://doi.org/10.1016/j.envsoft.2015.08.013.

---

## Author Response (AR1)

**Letter to Editor egusphere-2025-425**

We would like to thank the editor for managing the review process and the referees for their constructive comments, which significantly improved our manuscript.

Following the suggestions from Referee #2 (please refer to the discussion 'Reply on RC2'), we implemented a grid-search-based hyperparameter tuning procedure for the stand-alone LSTM used in our study. Specifically, we optimized three hyperparameters: the number of hidden states, the initial learning rate, and the dropout rate (frequently identified as critical in DL models). Additionally, besides the stand-alone LSTM and the hybrid model, we calibrated 196 individual HBV models locally, one for each catchment analyzed in this study. The calibration approach and results for the HBV models are detailed and discussed in the newly added Appendix B. In response to Reviewer #1's recommendation, we explored two further strategies to address the saturation behavior of the LSTM under extreme scenarios. These strategies included modifying the loss function and developing a specialized LSTM variant (sLSTM). The methodological description and results from these tests are provided in the newly included Appendix D. Furthermore, we refined the methods section and expanded our discussion of results to integrate these modifications effectively. We also incorporated all minor suggestions provided by both reviewers.

Below, we summarize the reviewers' major comments and outline the specific manuscript revisions made in response. Reviewer comments are highlighted in blue, with our responses in black. Key revisions in the manuscript are marked in **bold**. For a detail response to the reviewer comments, we refer to the discussion in HESSD.

**Referee Comments: RC1, Basil Kraft:**

1. I see potential for making the study much more impactful by testing solutions for the saturation effect. The authors speculate a lot in the discussion, and I was a bit disappointed that these ideas were not tested. However, I respect the authors' decision to not include further analysis. I do not insist on further experiments, but I would find them very interesting.

Response: In the first submission, we discuss the results from larger LSTM networks trained on more diverse datasets. However, following the referee's suggestion, we tested two additional strategies to mitigate the saturation effect, namely, a modified loss function and the sLSTM architecture. **Appendix D** gives the description of methods and results from these models. We also adjusted our discussion.

2. The hybrid model receives 730 days of input, while the LSTM only receives 365 days. The authors argue that the hybrid model needs the additional data for spin-up of the internal states. This makes sense, but the LSTM also needs to initialize its states. If the LSTM is used in a many-to-many setup (where the prediction period corresponds to the input period), the warm-up period should be the same for both models.

Response: **Section 2.4 (Line 147-151)** of the revised manuscript includes an explanation of the chosen training approach (seq-to-one for the LSTM and seq-to-seq for the Hybrid) and the respective sequence lengths used for both the models.

3. Why does the hybrid model receive an additional input? I believe the comparison between the models would be fairer if both models received exactly the same input data. Please clarify.

Response: We modified **Section 2.4 (Line 152-157)** of the manuscript for a clearer explanation of the use of pet\_sim (mm/d) as an explicit input to the HBV components of the Hybrid model.

4. The synthetic precipitation data is extracted for a selection of catchments within a distance of 2.5 km. If I understand correctly, the model is fed with point observations, while it was trained on the average precipitation over the catchment area. This could be a source of error. Please discuss or clarify.

Response: We revised **Section 2.5 (Line 169-174)** of the manuscript to include a clarification with respect to the said source of error.

5. I would appreciate a brief analysis of the extreme values within the test data. How do the predictions compare to the observations? If you see the same saturation in the test data, this would strengthen your argument. It would also rule out that the experimental setup (changing

precipitation but not temperature, etc.) is the cause of the saturation. You mention this in the discussion (L365-370), and I suggest showing it instead of speculating.

Response: We modified **sections 3.1** (Line **227-230**) to include the RMSE for the predictions of the stand-alone LSTM for extreme events within the test data. **In section 3.3** (Line **295-297**) we talk about the LSTM saturation (%) for these events without the input of synthetic precipitation data.

**RC2: Anonymous:**

1) Even though it is outside the scope of the paper, I would have appreciated a "deeper" and "fairer" comparison between the stand-alone LSTM model and the hybrid HBV-LSTM model. The paper is short (only 3 figures of results in the main text), there is room for that. My main criticism is that no hyperparameter (HP) tuning is done for either model. The HP values are simply taken from previous studies. I think that the results could be different if a proper fine tuning was done for each model.

Fig 1: Validation NSE for grid search based hyperparameter tuning. The best model (in red) does not affect results of this study in comparison to the model originally used in the study (in blue).

Response: On the referee's suggestion, we implemented a grid search based hyperparameter tuning for our LSTM model. The following hyperparameters were searched for from among the given values, respectively:

1. Number of Hidden states: 64, 96, 128, 156, 196, 224, 256

2. Initial learning rate 0.001, 0.0005

3. Dropout rate: 0.0, 0.25, 0.4, 0.5

We trained a single model for every combination of the hyperparameters for the training period (01.10.1995 - 30.09.2005) and tested them for the validation period (01.10.2005 - 30.09.2010). The evolution of the validation performance of these models in terms of the CDF of the catchment-wise NSE is given in Fig. 1. The best model hyperparameter combination is given in red, which is a hidden size of 156, an initial learning rate of 0.001 and a dropout rate of 0.25. The validation performance of the hyperparameters used in this study is shown in blue. Though the slight difference between the two, might be significant for model benchmarking, it is not critical for the experimental set-up in our study. Training the LSTM with the best hyperparameter setting does not change the 'theoretical prediction limit' or the 'design limits' significantly, and hence does not change the nature of results in this study. Also, when we test ensemble models with differing hidden states (ranging from 8 to 2048, while keeping all other hyperparameters unchanged), it does not affect the nature of our results. Thus, we believe that, the results from the hyperparameter tuning or any inferences drawn from it can be left out from the revised manuscript. Instead, we stick to the adopted hyperparameters based on previous studies by Acuna Espinoza et al., (2024) and Kratzert et al., (2019). We also believe that it is justified to adopt the same hyperparameters of the LSTM for the Hybrid model, for a fair comparison.

2) For analysis, it would also be very interesting to see the results for a single HBV model as a benchmark, which is very "cheap" to calibrate locally. Is there an improvement and is it "worth" the huge amount of data and GPU time required to process it? For example, the authors added US CAMEL data to their CH CAMEL learning dataset and moved from 64 to 256 nodes, which would have required a considerable amount of additional resources, but they don't show the corresponding improvement.

Response: We locally calibrated a variant of the HBV (Seibert, J. (2005)) model for all the catchments in our study. We include the description of the calibration process and discuss the results **in Appendix B** in the revised manuscript.

3) As the paper focuses on extremes, I also think that the evaluation against the observed runoff should not be limited to the NSE criteria as in Fig. 1 (which is the only figure presenting models performances), but should include a deeper analysis, including for example signatures calculated on flood events.

Response: We added the metrics High Flow Bias, fraction of Missed Peaks and Peak Mean Absolute Percentage Error in **Fig. 1** of the revised manuscript and also modified the **section 3.1** (Line 220-227).

4) The same comment applies to the second part of results (Figs. 2 and 3, using synthetic rainfall): only 1 flood for 3 catchments (a little more in the appendix), whereas the authors have thousands of examples. A synthetic metric should be found that "summarises" the different observed behaviours (between catchments, but also for the same catchment but under different conditions). A "visual" analysis on a few examples, as in this paper, is a first step to draw first hypotheses. But then these hypotheses should be tested in depth.

Response: We modified **section 3.2 (268-269, 277-280, 281-285)** to include a clarification of why we show only three flood events in Fig. 2. We added additional results in **Appendix B.** Regarding the reviewer's suggestion of developing a synthetic metric: we believe that at this stage, developing such a metric is best left as a part of future work, since it can be a research line on its own.

5) This last point (the need for a synthetic metrics that allows a "deep" analysis) leads me to my main comment. The authors don't clearly explain why, from a hydrological point of view, peak discharge should increase linearly with extreme rainfall. I fully agree with this, and even if it seems obvious, I think it would be valuable to anchor the paper with more basic hydrological references. In terms of synthetic metrics, I would, for example, calculate a regression coefficient between peak discharge and synthetic rainfall and see how it changes as a function of rainfall, as

in the paper, but also as a function of the initial moisture content before a flood and/or the runoff coefficient. I would also not look at flood by flood, but try to find a graphical representation of all floods and catchments together.

Response: The sensitivity of flood peaks to an increase in maximum precipitation varies significantly across catchments, depending on multiple factors such as topography, soil characteristics, land use, and antecedent moisture conditions (as correctly highlighted by the reviewer). For instance, Froidevaux et al. (2015) found that 0–3 days of accumulated precipitation is the main driver of floods, while longer-term (4 days–1 month) antecedent precipitation has only weak, region-specific effects—especially relevant in gentler plateau areas, but negligible in Alpine catchments. While Staudinger et al. (2025) found that only 18–44% of extreme annual floods coincided with maximum precipitation, highlighting the crucial role of antecedent soil moisture and snow storage. Several attempts from our side showed that given these complexities, it is challenging to identify a consistent, clear signal across a large-sample dataset covering Switzerland, with its diverse hydrological regimes. We have improved our manuscript in this regard. We modified section 4 (line 407-422) of the manuscript by clearly articulating, from a hydrological perspective, what physically reasonable runoff responses should look like, and explicitly discuss the limitations observed in the LSTM predictions for extreme events.

**References:**

- Acuña Espinoza, Eduardo, Ralf Loritz, Manuel Álvarez Chaves, Nicole Bäuerle, and Uwe Ehret. 2024. "To Bucket or Not to Bucket? Analyzing the Performance and Interpretability of Hydrological Models with Dynamic Parameterization." *Hydrology and Earth System Sciences* 28 (12): 2705–19. https://doi.org/10.5194/hess-28-2705-2024.
- Froidevaux, P., J. Schwanbeck, R. Weingartner, C. Chevalier, and O. Martius. 2015. "Flood Triggering in Switzerland: The Role of Daily to Monthly Preceding Precipitation." *Hydrology and Earth System Sciences* 19 (9): 3903–24. <a href="https://doi.org/10.5194/hess-19-3903-2015">https://doi.org/10.5194/hess-19-3903-2015</a>.
- Kratzert, Frederik, Daniel Klotz, Guy Shalev, Günter Klambauer, Sepp Hochreiter, and Grey Nearing. 2019. "Towards Learning Universal, Regional, and Local Hydrological Behaviors via Machine Learning Applied to Large-Sample Datasets." *Hydrology and Earth System Sciences* 23 (12): 5089–5110. https://doi.org/10.5194/hess-23-5089-2019.
- Seibert, J. (2005) HBV Light Version 2. User's Manual. Department of Physical Geography and Quaternary Geology, Stockholm University, Stockholm.
- Staudinger, Maria, Martina Kauzlaric, Alexandre Mas, Guillaume Evin, Benoit Hingray, and Daniel Viviroli. 2025. "The Role of Antecedent Conditions in Translating Precipitation Events into Extreme Floods at the Catchment Scale and in a Large-Basin Context."

  Natural Hazards and Earth System Sciences 25 (1): 247–65. https://doi.org/10.5194/nhess-25-247-2025.

---

## Referee Report (RR1)

**"Unveiling the Limits of Deep Learning Models in Hydrological Extrapolation Tasks"**

Manuscript #2025-425; Second revision

B. Kraft

July 28, 2025

**General feedback**

The authors have addressed the reviewer's comments in a satisfactory and thoughtful manner. The revisions improve the clarity and rigor of the manuscript, and the responses to the concerns raised during the review process are appropriate and well-reasoned.

A few minor issues remain, mostly related to clarity and phrasing, as detailed below. I recomment to address these issues before publication, but they do not significantly affect the overall quality of the manuscript. Therefore, I suggest accepting the manuscript subject to technical corrections.

**Minor remarks**

Here I list some typos and suggestions for improving clarity. Line numbers refer to the diff PDF.

- L5 I wonder whether hybrid models are known in the community or if a brief explanation of the term would help. Probably "hybrid process/machine learning model"?
- L9 CAMELS-CH is mentioned here for the first time, but not defined. I suggest "below the maximum value of  $183 \text{ mm d}^{-1}$  in the training data.", and suggest not mentioning CAMELS-CH here.
- L121 Would the maximum for the combined dataset not be the 299 mm  $d^{-1}$  from the US dataset?
- L375 "but accompanied by a decrease in overall performance" instead of "by a loss"?
- L376 "Both strategies ..." instead of "Both the strategies ...".
- L377 Not sure if this is a typo, do you really believe that these issues can be resolved completely? Maybe you meant "cannot be resolved completely"? In both cases, I suggest rephrasing, either tone down or further elaborate why this can be resolved completely.
- L429 "This suggests ..." instead of "It is thereby obvious ..."?
- L433 Sentence seems to be incomplete. "While  $\dots$ " should be followed by a contrastive statement.
- L420-442 This part would benefit from a rewrite to enhance clarity and readability.
  - L455 You could highlight that the hybrid model, by following the physical constraints, is also biased in a way towards prior knowledge and assumptions.